# Secretome from Magnetically Stimulated Muscle Exhibits Anticancer Potency: Novel Preconditioning Methodology Highlighting HTRA1 Action

**DOI:** 10.3390/cells13050460

**Published:** 2024-03-05

**Authors:** Yee Kit Tai, Jan Nikolas Iversen, Karen Ka Wing Chan, Charlene Hui Hua Fong, Rafhanah Banu Abdul Razar, Sharanya Ramanan, Lye Yee Jasmine Yap, Jocelyn Naixin Yin, Shi Jie Toh, Craig Jun Kit Wong, Pei Fern Angele Koh, Ruby Yun Ju Huang, Alfredo Franco-Obregón

**Affiliations:** 1Department of Surgery, Yong Loo Lin School of Medicine, National University of Singapore, Singapore 119228, Singapore; nikolas.iversen@u.nus.edu (J.N.I.); surwjkc@nus.edu.sg (C.J.K.W.); 2Institute of Health Technology and Innovation (iHealthtech), National University of Singapore, Singapore 117599, Singapore; 3Biolonic Currents Electromagnetic Pulsing Systems Laboratory (BICEPS), National University of Singapore, Singapore 117599, Singapore; 4NUS Centre for Cancer Research, Yong Loo Lin School of Medicine, National University of Singapore, Singapore 117599, Singapore; 5Cancer Science Institute of Singapore, National University of Singapore, Singapore 117599, Singaporerubyhuang@ntu.edu.tw (R.Y.J.H.); 6Department of Obstetrics & Gynaecology, National University of Singapore, Singapore 119228, Singapore; 7Graduate Institute of Oncology, College of Medicine, National Taiwan University, Taipei 10617, Taiwan; 8School of Medicine, College of Medicine, National Taiwan University, Taipei 10617, Taiwan; 9Healthy Longevity Translational Research Programme, Yong Loo Lin School of Medicine, National University of Singapore, Singapore 119228, Singapore; 10Department of Physiology, Yong Loo Lin School of Medicine, National University of Singapore, Singapore 117593, Singapore; 11Competence Center for Applied Biotechnology and Molecular Medicine, University of Zürich, 8057 Zürich, Switzerland

**Keywords:** exercise, breast cancer, myokines, exerkines, proteomics, PEMFs, conditioned media, soluble biomarkers, metastasis, aminoglycoside antibiotics

## Abstract

Briefly (10 min) exposing C2C12 myotubes to low amplitude (1.5 mT) pulsed electromagnetic fields (PEMFs) generated a conditioned media (pCM) that was capable of mitigating breast cancer cell growth, migration, and invasiveness in vitro, whereas the conditioned media harvested from unexposed myotubes, representing constitutively released secretome (cCM), was less effective. Administering pCM to breast cancer microtumors engrafted onto the chorioallantoic membrane of chicken eggs reduced tumor volume and vascularity. Blood serum collected from PEMF-exposed or exercised mice allayed breast cancer cell growth, migration, and invasiveness. A secretome preconditioning methodology is presented that accentuates the graded anticancer potencies of both the cCM and pCM harvested from myotubes, demonstrating an adaptive response to pCM administered during early myogenesis that emulated secretome-based exercise adaptations observed in vivo. HTRA1 was shown to be upregulated in pCM and was demonstrated to be necessary and sufficient for the anticancer potency of the pCM; recombinant HTRA1 added to basal media recapitulated the anticancer effects of pCM and antibody-based absorption of HTRA1 from pCM precluded its anticancer effects. Brief and non-invasive PEMF stimulation may represent a method to commandeer the secretome response of muscle, both in vitro and in vivo, for clinical exploitation in breast and other cancers.

## 1. Introduction

Skeletal muscle accounts for approximately 40% of the total body mass of a healthy adult [1]. Muscle metabolism is linked to overall human health [2] and can be enhanced with exercise [3,4]. In particular, aerobic exercise augments acute and basal fat utilization, thereby attenuating systemic inflammation, which promotes metabolic stabilization and resilience to disease. Aerobic exercise, as it relies heavily on muscular mitochondrial respiration, spurs muscle adaptation towards a more oxidative phenotype. Oxidative muscles are characterized by increased mitochondrial density, improved antioxidant capacities, and a substrate preference for fatty acid β-oxidation for energy production [3]. The mitochondrial-associated transcriptional pathways recruited during oxidative muscle determination are upstream of the secretome response of muscle [5,6] which serve at a systemic level to increase basal metabolic rate, improve insulin sensitivity, and foster fat utilization [3,7,8].

Exercise reduces the risks of developing breast [9], prostate [10], and colon [11] cancers. Exercise, undertaken before, during, or after cancer therapy, is associated with improved human breast cancer survival [9,12]. Exercise in animals has also been shown to decrease cancer incidence, tumor growth, and aggressiveness [13,14]. The beneficial effects of exercise over cancer are due to the production and release of anticancer myokines from activated muscles [4,15,16]. As a manifestation of the exercise-induced muscle secretome, blood collected from mice [14], healthy individuals [17], and prostate [10] or breast [18] cancer patients having exercised inhibited the growth of prostate or breast cancer cells in vitro. Blood harvested from individuals who had undertaken moderate to high-intensity exercise exhibited the greatest anticancer potency, which may not be realizable in many patients [10,12,17]. Despite the clear benefits of exercise for cancer patients, compliance and adoption have proven challenging [19] and would be exceptionally difficult in patients whose functional status has deteriorated greatly due to disease progression and treatment-related side effects.

The systemic metabolic adaptations produced by exercise are orchestrated by the muscle secretome. Muscle myokines mediate the trophic interactions between muscle and the rest of the body’s other tissues [20]. The myokine pool consists of a collection of cytokines and other bioactive agents that, following their release into the interstitial fluid and bloodstream, interact with muscle and other tissues to modulate their metabolism and inflammatory status. Certain exercise-induced myokines (exerkines) are known to stymie cancer development and progression by inhibiting malignant proliferation, epithelial-mesenchymal (EMT) transition, migration, and invasion [7,15,16,21,22] as well as make the tumor microenvironment less hospitable for cancer progression [21]. Myokines mitigate cancer-permissive comorbidities such as obesity, low-grade inflammation, and dysfunctional immune responses [7,20].

The unmet need was a non-strenuous manner to safely invoke the anticancer attributes of the muscle secretome in patients too frail to exercise. Brief exposure to pulsed electromagnetic fields (PEMFs) was previously shown to promote myogenesis in vitro and in vivo [23,24]. These low-energy magnetic fields were shown to activate the PGC-1α (peroxisome proliferator-activated receptor-gamma coactivator (PGC)-1alpha) transcriptional pathway [3,23,24], classically associated with exercise-induced oxidative adaptations characterized by increased muscular mitochondriogenesis, enhanced systemic fatty acid oxidation, and improved insulin sensitivity [5,6]. Accordingly, the PEMF paradigm employed in the present study has been previously shown to produce parallel mitohormetic adaptations in cells [3,23,24,25], animals [23], and humans [26,27]. Most relevantly, as the myokine response is PGC-1α-dependent [5,6,24], it can be activated by brief PEMF exposure [23,24,25].

High-Temperature Requirement A1 (HTRA1) is a candidate anticancer myokine [28]. HTRA1 was previously shown to be upregulated by direct exposure to PEMFs as well as in response to the administration of PEMF-induced conditioned media, pCM [25]. HTRA1 levels are downregulated in pancreatic, gastric, prostate, lung, bladder, hepatocellular carcinoma, ovarian, and breast cancers and correlate with increased tumor growth, epithelial-mesenchymal transition (EMT), metastasis, and chemoresistance [29,30]. Depressed levels of HTRA1 are correlated with more aggressive forms of breast cancers characterized by nodal metastasis, estrogen receptor downregulation, high-grade tumors exhibiting p53 point mutations, lymphocytic infiltration [31], and reduced disease-free survival [29,31]. HTRA1 is predominantly secreted into the bloodstream as a cleavage product of the full-length protein, conferring upon it paracrine action on adjacent and distal tissues [29]. HTRA1 secretion is also correlated with inflammatory states, which would be expected if it were implicated in mitohormetic adaptive responses [32]. The scenario may be analogous to that of IL-6, which is interpreted as being anti-inflammatory when released by exercise, yet proinflammatory when associated with states of obesity and insulin-resistance [33]. Serum HTRA1 levels are hence elevated in patients with age-related macular degeneration [34] as well as pregnancy-related hypertension (preeclampsia) [35]. In accordance with a putative role in mitohormetic responses, muscular HTRA1 expression is driven by oxidative stress [25,28,36], where it contributes to muscle development [37]. While HTRA1 induction with exercise has yet to be convincingly shown, HTRA1 expression has been shown to be upregulated upon magnetic exposure of muscle cells [25]. An open question is whether HTRA1 underlies similar anticancer responses following PEMF therapy and exercise.

This study was undertaken to investigate the potential of PEMF therapy to serve as a non-invasive manner to efficiently mobilize the secretome of muscle for soluble biomarker discovery and anticancer validation. The ability of muscle cell pCM and blood serum from PEMF-exposed mice to suppress breast cancer cell growth, migration, and invasion was investigated in vitro using several breast cancer cell models. The anticancer potency of pCM was also examined on breast cancer microtumors implanted onto the chicken egg chorioallantoic membrane (CAM). A method of further enhancing the anticancer potency of the pCM was developed employing directionally specific magnetic field exposure in conjunction with pCM preconditioning. Notably, HTRA1 levels were elevated within the pCM of muscle cells and were necessary for myogenic progression. HTRA1 levels were also upregulated in the blood of exercised and PEMF-exposed mice, substantiating potential mechanistic commonalities between PEMF exposure and exercise and providing physiological relevance to the preconditioning paradigm. This study sought to examine the contribution of HTRA1 to the anticancer efficacy of the muscle secretome.

## 2. Materials and Methods

### 2.1. Cell Culture and Chemical Reagents

The culturing of MCF-7 (HTB-22™) was previously described [38]. MDA-MB-231 cells were obtained from Dr. Glenn Kunnath Bonney, NUS, and maintained in Dulbecco’s Modified Eagle Medium (DMEM) (Hyclone; Danaher, Washington, DC, USA) with 10% FBS. Murine 4T1 and 67NR breast cancer cells were acquired from Dr. Lina Lim, NUS, and were adapted to grow in DMEM with 10% FBS. No cell culture antibiotics were used throughout the experiments. Recombinant HTRA1 was purchased from Cusabio (CSB-EP8795; recombinant Mouse HTRA1 from *Escherichia coli*).

### 2.2. Generation of Muscle-Conditioned Media

The culturing of C2C12 cells was previously described [25]. Briefly, differentiated myotubes in serum-free DMEM were subjected to 1.5 mT PEMF for 10 min and allowed to repose for 6 h or 24 h in a standard cell culture incubator. Control cells were placed in the PEMF device without powering. Conditioned media was collected after centrifugation at 1200 rpm for 5 min.

### 2.3. Cancer Cell Count, Colony Formation, Invasion and Migration Assays

Cell enumeration was performed using trypan blue exclusion assay, colony formation or DNA content determination (CyQuant Cell Proliferation Assay kit, Thermo Fisher Scientific, Waltham, MA, USA) as previously described [25,38].

Briefly, muscle-conditioned media were provisioned to breast cancer cells 24 h post seeding in a 6-well plate format, in 3 technical replicates per condition. Cell count using trypan blue exclusion assay was performed 24 h post muscle-conditioned media incubation. For CyQuant DNA content analysis, breast cancer cells, seeded in a 96-well format with 8 technical replicates per condition, were similarly treated and analyzed after the provision of muscle-conditioned media. CyQuant stained DNA was measured at 480/520 nm using a Cytation 5 microplate reader (BioTek, Winooski, VT, USA).

In vitro colony formation assay was performed using crystal violet staining. Briefly, MCF-7 or 4T1 breast cancer cells were seeded at 100 per well of a 6-well plate (day 0). MCF-7 cells were treated with muscle-conditioned media on day 1, day 4 and day 7 while 4T1 cells were treated on day 1 and day 4. MCF-7 and 4T1 cells were rinsed in PBS and stained with crystal violet stain [38] on day 11 and day 7, respectively. Images of the colonies were taken using ChemiDoc Imaging System (Bio-Rad, Hercules, CA, USA) under the Coomassie Blue Stain filter setting. The number of colonies per well was estimated using the ImageJ (Version 1.53, U.S. National Institutes of Health, Bethesda, MD, USA) Analyze Particle option and was expressed as fold change over control.

The invasion and migration capacities of breast cancer cells were also previously described [38]. Briefly, MCF-7 or 67NR cells at a density of 30,000 cells were seeded into each quadrant of a 4-well 3.5 mm culture dish insert (ibidi) in their respective media. The cells were left to adhere for 24 h before the removal of the insert and seeding media before the addition of muscle-conditioned media to a total volume of 2 mL per dish. Closure of the gaps was captured using light microscopy on all 4 limbs of the insert, taken every 24 h. The average of 32 gap distances was considered from the 4 limbs with 8 readings arising from each limb. The images of the gap distances were analyzed using ImageJ.

Invasion assay was performed using the CytoSelect 24-well Cell Invasion Assay kit (Cell Biolabs, Inc., San Diego, CA, USA) according to the manufacturer’s protocol. Briefly, MDA-MB-231 or 4TI breast cancer cells (195,000 cells) were seeded in the cell culture insert after the rehydration of the basal membrane in FBS-free media. The lower well of the invasion plate was filled with their muscle-conditioned media supplemented with 5% FBS to promote the invasion of cells through the basal membrane. The setup was incubated for 48 h in a standard tissue culture incubator before the extraction and staining of the invaded cells from the basal membrane. The lysates from the extracted cells were analyzed at OD 560 using a Cytation 5 microplate reader (BioTek).

### 2.4. HTRA1-Depletion Conditioned Media

HTRA1 in muscle CM was depleted using an antibody and agarose bead pulldown kit. Briefly, 3 mL of CM was preincubated with 100 µL of protein A/G beads (Santa Cruz, Santa Cruz, CA, USA, SC-2003) for 1 h. After a spin of 1000× *g* for 5 min, the CM was incubated with 5 µg of HTRA1 antibody (Proteintech, Rosemont, IL, USA, 55011-1-AP) or control rabbit IgG (Vector Laboratories, Newark, CA, USA, I-1000-5) overnight. A volume of 65 µL of protein A/G was then added to the CM for 2 h, followed by a 1000× *g* spin for 1 min. Further cleansing of the CM was repeated with 65 µL protein A/G for an additional 1 h incubation and 1000× *g* spin for 1 min. Lastly, the HTRA1-depleted CM was passed through a 0.45 µM filter (Merck, Rahway, NJ, USA, SLHPR33RS) and administered to 67NR cells for cell count analysis.

### 2.5. Chicken Chorioallantoic Membrane (CAM) Model and Ultrasound Assessment

The CAM model was performed based on published protocol [38]. For an extended protocol, refer to Appendix A. Briefly, embryonic day 0 (ED 0) fertilized Bovan Goldline Brown chicken eggs from Lian Wah Hang Pte Ltd. (Singapore) were placed in a 38.5 °C incubator with 70% humidity (Rcom MX-50, AUTOELEX Co., Ltd., Gyeongsangnam-do, Republic of Korea). On ED 3, some albumin was removed and a 1 cm^2^ window on the shell was created. On ED 6, pelleted MCF-7 cells in Matrigel (Corning, New York, NY, USA) were pipetted onto the micropunctured CAM vessel. Delivery of muscle CM was conducted on a filter paper adjacent to the microtumors. Microtumors were treated with fresh muscle CM for 4 days.

### 2.6. Myotube Differentiation and Preconditioning Paradigm

The collection of magnetically induced conditioned media (CM) from C2C12 suspension cells was previously described [25]. Briefly, myoblasts suspended in fresh DMEM were exposed to 1.5 mT PEMFs for 10 min in the downward (pCM, down) or upward (pCM, up) direction to produce preconditioning media. cCM (constitutive release, non-PEMF) was produced from unexposed myoblast suspension cells. After 1 h of post-magnetic stimulation, isolated cCM or pCM from myoblasts were supplemented with 2% horse serum (Hyclone; Thermo Fisher Scientific) to generate differentiation media for early differentiating myoblast sister cultures. These differentiating cultures were given fresh differentiation media every 2 days before the analysis of the whole cell lysates and conditioned media using Western blot or ELISA.

### 2.7. Western Blot and Silencing of HTRA1

Western blot was performed according to published protocols [23,25]. Briefly, cell lysates were prepared in RIPA buffer and 25–50 µg of total protein or crude CM was resolved using 8–12% denaturing PAGE gel and transferred to PVDF membrane (Immobilon-P, PVDF).

The primary antibodies used are listed in Table 1.

The membranes were visualized using SuperSignal West Pico or West Femto chemiluminescent substrate (Thermo Fisher Scientific) on the LI-COR Image Studio (LI-Cor Odyssey FC). Silencing of HTRA1 (NM_019564) in C2C12 myoblasts was performed using 2 independent dicer-substrate interfering RNAs (dsiRNA). The 2 dsiRNAs target the coding sequence of exon 4 and 5 (dsi1) and 3′-untranslated region of exon 9 (dsi2). The dsiRNAs were acquired from Integrated DNA Technologies (Coralville, IA, USA). Transfection of dsiRNAs at 10 nM was carried out using Lipofectamine 2000 (Thermo Fisher Scientific) according to the manufacturer’s instructions. Briefly, subconfluent C2C12 myoblasts were transfected with dsiRNA for 3 consecutive days before being given differentiation media supplemented with 5% horse serum 24 h after the last transfection. Analysis of myogenic proteins was performed 3 days after the provision of differentiation media.

### 2.8. Animal Study Protocol and Serum ELISA

The animal study (R21-0213) was approved by the Institutional Animal Care and Use Committee (IACUC, NUS). Animal experiments were performed on 7-week-old female C57BL/6J mice. Acclimatized mice were subjected to either control, PEMF, or exercise interventions for 8 weeks as previously described [23]. For an extended protocol, refer to Appendix A. The abundance of HTRA1 in animal sera was measured using a mouse HTRA1 kit (abx519097, Abbexa, Houston, TX, USA) and was performed according to the manufacturer’s instructions.

### 2.9. Statistical Analyses

All statistics were analyzed using GraphPad Prism (Version 10.2.0 for Windows, GraphPad Software, Boston, MA, USA). One-way analysis of variance (ANOVA) was used to compare the values between two or more groups supported by multiple comparisons. This was followed by Bonferroni’s post hoc test. A two-tailed *t*-test was used for comparisons between the mean of two independent samples.

## 3. Results

### 3.1. PEMF-Conditioned Media (pCM) Inhibits Breast Cancer Growth In Vitro and Ex Vivo

A single 10 min exposure of myoblasts to 1.5 mT amplitude PEMFs in the downward direction was previously shown to generate a conditioned media (pCM) that best enhanced myogenesis [25]. We investigated whether the same PEMF exposure regime could stimulate the production and release of anticancer agents (Figure 1A). To this end, differentiated myotube cultures were exposed to 1.5 mT downward PEMFs for 10 min and allowed to condition the bathing media for 6 h before collecting the pCM. Henceforth, all PEMF exposures used for either the generation of pCM or for mouse blood serum (PEMF-Serum) will be at an amplitude of 1.5 mT applied for 10 min and given to cells or mice in the downward direction and compared to the conditioned media generated from control (unexposed) muscle cells (cCM) or unexposed mice (Control-Serum), unless explicitly stated otherwise.

The ability of murine muscle pCM to modulate the growth, invasion, and migration capacities of human and mouse breast cancer cells was first tested in vitro. Administering murine pCM to human (MCF-7; Figure 1B,C; blue) and mouse (4T1; Figure 1D,E; blue) cancer cells mitigated their capacity to form colonies by ~24% and ~49%, relative to basal media (BM, grey); cCM did not affect breast cancer cell colony formation. Invasiveness was investigated using the more invasive human MDA-MB-231 and mouse 4T1 breast cancer cell lines [38,39]. Murine pCM inhibited the invasion of both human MDA-MB-231 and murine 4T1 cancer cells, reflected by a reduction in the number of blue-stained invading cells (Figure 1F,H), by ~32% compared to basal media (BM) (Figure 1G,I; blue vs. grey). Notably, murine cCM administered to murine 4T1 cells reduced invasion by ~12% (Figure 1I; red vs. grey), whereas no difference was observed over human MDA-MB-231 cells (Figure 1G; red vs. grey). The migratory capacities of human (MCF-7; Figure 1J,K) and murine (67NR; Figure 1L,M) breast cancer cell lines were assessed while in the presence of BM, cCM, or pCM. Although the provision of murine pCM slowed gap closure for both cancer cell lines, statistical significance was only achieved with the murine cell line (67NR; Figure 1M), recapitulating the species specificity detected with the invasion assay.

cCM and pCM were next tested on MCF-7 breast cancer microtumors hosted by the chorioallantoic membrane (CAM) of fertilized chicken eggs. The CAM is a proven vascular membrane model for studying tumor growth and angiogenesis [38,40]. MCF-7 microtumors were administered daily (Figure 2A; triangles) either 1× or 10× concentrated pCMs or cCMs (Figure 2B) over four days. The provision of 10× concentrated pCM (hatched blue; Figure 2C) resulted in a significant reduction in MCF-7 tumor volume as compared to 10× cCM (hatched red; Figure 2C). Moreover, the assessment of tumor vascularity by 3D ultrasound showed a significant attenuation of tumoral blood vessel formation in the microtumors of both 1× (solid blue) and 10× concentrated pCM (hatched blue) (Figure 2D), relative to 1× or 10× concentrated cCMs, respectively.

### 3.2. PEMF Treatment of Mice Produces Anticancer Blood Serum

To validate PEMF-induction of the muscle secretome at a higher organismal level, normal mice were exposed to PEMFs once a week for 8 weeks, followed by blood collection one week after the last PEMF exposure (Figure 3A). The efficacy of the PEMF-induced sera was compared against sera collected from exercised mice for the same 8-week intervention and collection (9th week) protocol. Both PEMF-serum (blue) and exercise-serum (green) significantly reduced the colony-forming capacity of murine 4T1 breast cancer cells (Figure 3B,C) as well as significantly reduced murine 67NR cellular DNA content (Figure 3D) compared to control-serum (red), indicating depressed cancer cell proliferation in response to the treatment. Furthermore, the PEMF-serum (blue) reduced the invasion of breast cells relative to both exercise-serum (green) and control-serum (red) and was more pronounced for the murine 4T1 (Figure 3E,F) than for human MDA-MB-231 (Figure 3G,H) cells. Finally, both the PEMF- (blue) and exercise- (green) sera inhibited the migration of 67NR cells (Figure 3I,J), whereas only the PEMF-serum (blue) reduced the migration of human MCF-7 cells, relative to control-serum (red) (Figure 3K,L).

### 3.3. Preconditioning Paradigm to Accentuate the Anticancer Potency of pCM

The finding that the mouse sera collected one week after the last PEMF exposure or exercise session exhibited anticancer potency implied an adaptive response to the interventions (Figure 3). One manner in which adaptation may have been instilled is via the actions of the muscle secretome itself. To ascertain whether a history of pCM treatment could render adaptations that would later translate to augmented basal or stimulated secretome responses, myoblast pCM was used as a preconditioning stimulus during the early stages of in vitro differentiation (Figure 4A). To this end, proliferating myoblasts were preconditioned with either fresh basal media (grey), cCM (red), or pCM, generated in response to either upward (hatched blue) or downward (blue) magnetic field exposure of myoblasts while in suspension (Figure 4B) [25]. The outcome measure was the ability of the resultant murine myotube PEMF-induced secretome (following preconditioning) to retard the growth of the murine 67NR breast cancer cell line. In all cases, the collection of secretome from myotubes in basal media (grey bar) gave similar secretome responses to their preconditioning with cCM (red). Moreover, preconditioning with cCM, or no preconditioning with basal media, produced indistinguishable myotube secretomes whether the comparison was made in the absence (Figure 4B, light grey shaded region) or presence (Figure 4B, dark grey shaded region) of PEMF exposure. This result indicates that the preconditioning of muscle cells with constitutively released secretome (cCM) from unexposed myoblasts nominally influences conditioned media composition, being similar in response to fresh basal media.

By contrast, the preconditioning with secretome from PEMF-treated myoblasts (down fields, solid blue; up fields, hatched blue) enhanced the anticancer potency of the secretome generated from the resulting myotubes both in the absence (Figure 4B, light grey shaded background) and presence (Figure 4B, dark grey shaded background) of PEMF exposure; the effect being greatest from PEMF-exposed myotubes (dark grey shaded background) preconditioned with myoblast downward pCM (rightmost solid blue bar). pCM preconditioning (down more than up) thus augmented both constitutive (light grey shaded background) as well as PEMF-triggered (dark grey shaded background) secretome release from myotubes.

To better illustrate that preconditioning with pCM served as an adaptive stimulus that later enhanced both constitutive as well as PEMF-induced secretome release, each preconditioning paradigm was compared to itself in the PEMF exposed (+) and unexposed (−) conditions (Figure 4C). PEMF exposure (+) consistently enhanced the anticancer efficacy of the generated myotube pCM. However, constitutive release (−) was only enhanced in response to preconditioning with pCM generated from upwards (hatched blue) or downwards (solid blue) exposure of myoblasts. The greatest response was achieved with direct myotube exposure to downward PEMFs preceded by preconditioning with the downward field-generated pCM from myoblasts (solid blue, second bar).

We next specifically examined the effects of downward field-generated secretome preconditioning on PEMF-induced secretome release. The secretomes generated from pCM (down) preconditioned myotubes, with or without direct myotube PEMF exposure (blue bars), were tested in colony formation, invasion, and migration assays and were compared against the effects of the constitutively released secretome (without PEMF exposure) harvested from cCM-preconditioned myotubes (red bar) (Figure 5A). These conditions correspond to the black triangles shown in Figure 4B. Murine myotube secretome preconditioned with pCM alone (without direct myotube PEMF exposure) suppressed human (MCF-7; Figure 5B,C; middle blue bar) and murine (4T1; Figure 5D,F; middle blue bar) colony formation by ~−19% and ~−13%, respectively. However, when pCM-preconditioned myotubes were additionally exposed to PEMFs (right blue bar), the resulting myotube secretome suppressed colony formation by −38% and −37% for MCF-7 and 4T1 cells, respectively. The invasiveness of both human (MDA-MB-231; Figure 5F,G) and murine (4T1; Figure 5H,I) cells was suppressed by ~−8% and ~−7%, respectively, by pCM-conditioned myotube secretome alone (Figure 5G,I; middle blue bar), whereas the addition of direct exposure to myotubes enhanced the inhibitory capacity of the myotube secretome to ~−23% for both cell lines (right blue bar). Finally, with reference to cancer cell migration, the PEMF-induced secretome from pCM-preconditioned myotubes produced the strongest slowing of migration for both human (MCF-7; Figure 5J,K; blue circles) and murine (67NR; Figure 5L,M; blue circles) cancer cell lines. Previously, the constitutively released secretome from murine myotubes was shown to give nominal results in the human breast cancer cells (Figure 1K). Notably, the constitutively released secretome from pCM-preconditioned myotubes slowed the migration of both human (MCF-7; Figure 5J,K; blue squares) and murine (67NR; Figure 5L,M; blue squares) cells, relative to cCM-preconditioning alone (red squares). A drug-free and non-invasive in vitro method was thus established with which to enhance the anticancer breadth and potency of the muscle secretome.

### 3.4. HTRA1 Instigates Preconditioning Efficacy

A myokine basis for the adaptive pCM preconditioning response was next sought. HTRA1 expression was previously shown to be upregulated in C2C12 muscle cells by PEMF exposure [25]. Here, we show that HTRA1 was also detected in the conditioned media of adherent myotube cultures 24 h following PEMF exposure, but failed to reach detection levels from adherent myoblast cultures over the same period of conditioning (Figure 6A and Appendix A). On the other hand, HTRA1 could be detected in the pCM of myoblasts that were exposed while in suspension and then allowed to condition the media for 1 h (Figure 6B) [25]. Aligning with existing reports that HTRA1 is required for myogenic progression [37], HTRA1 levels were upregulated during myogenic differentiation (Figure 6C), whereas silencing HTRA1 expression impeded myogenic differentiation (Figure 6D). Accordingly, HTRA1 protein levels were correlated with those of MyoD (Myogenic Differentiation 1) and p21 (wildtype activating factor-1/cyclin-dependent kinase inhibitory protein-1 or CIP1/WAF1) [41] during myogenic differentiation (Figure 6C) and the genetic silencing of HTRA1 (Figure 6D). Moreover, as HTRA1 expression is upregulated by pCM administration [25], the preconditioning of early muscle cultures with HTRA1-enriched pCM generated from myoblast suspensions (Figure 6B) further enhanced both HTRA1 expression as well as myogenic differentiation (Figure 6E) that should, in turn, be reflected as an increase in anticancer potency [28]. Specifically, the greatest increases in the protein expressions of MyoD, myogenin, desmin, and HTRA1 were achieved with direct exposure of myotubes to downward PEMFs preceded by preconditioning with pCM generated from myoblasts in suspension exposed to downward PEMFs (rightmost solid blue bars in (ii), (iii), (iv), and (v)). This pCM preconditioning arrangement (Figure 4, Figure 5, Figure 6) also most strongly increased the anticancer potency of the myotube secretome over constitutive release, depicted by the grey triangles in Figure 6E (basal media, or no preconditioning or PEMF exposure). Notably, optimal secretome responses occurred with preconditioning with pCM generated from myoblasts in suspension with media conditioning of one hour, instead of myotube pCM or pCM collected from adherent myoblasts for longer conditioning times (Appendix A). This demonstrated the myoblast pCM-preconditioning paradigm thus serves as a manner to enhance the secretome response of muscle for liquid biomarker discovery and characterization (Figure 6F).

HTRA1 expression thus developmentally recapitulated key features of our preconditioning paradigm. Demonstrating necessity and sufficiency, recombinant HTRA1 was capable of attenuating MCF-7 growth (Figure 6G), whereas antibody absorption of HTRA1 from myotube pCM abolished anticancer efficacy (Figure 6H). Finally, providing physiological relevance to our in vitro preconditioning paradigm, HTRA1 was found to be upregulated in the sera of PEMF-exposed and exercised mice (Figure 6I), aligning with previous data that analogous PEMF exposure promotes both in vitro [24] and in vivo [23] myogenesis. Although, based on these experiments, the anticancer potency of the muscle cell secretome correlates closely with HTRA1 expression, it cannot, at this juncture, be considered exclusive.

## 4. Discussion

Regular exercise is proven to benefit health and lifespan [20]. Even so, the healthful benefits of exercise are under-exploited in modern society due to lifestyle restrictions. Exercise is particularly burdensome, if not impossible, for the infirm and frail, where the benefits of exercise are most needed. Exercise reduces the risk of developing cancer as well as improves the chances of surviving cancer by hindering the onset and progression of cancer and of cancer-associated inflammatory disorders [8]. Mechanistically, exercise-induced mitochondrial respiratory activation sets off a chain of events that culminate in the release of muscle secretome factors into the systemic circulation that are essential in disease management [42], including cancer [4].

Non-invasive, low-energy, and directionality-specified PEMFs have been previously shown to promote myogenesis via the activation of a mitochondrial–secretome signaling axis that is antagonized by aminoglycoside antibiotics [24,25]. Here, it is demonstrated that the same magnetically-induced secretome also possesses anticancer potency. The fact that both anticancer and promyogenic responses are antagonized by the aminoglycoside antibiotics indicates that they are synonymous (Appendix A). It is thus pertinent that HTRA1 is both promyogenic (Figure 6C–E) and anticancer (Figure 6G,H) and quickly released by PEMF exposure as a general feature.

This same PEMF exposure paradigm was previously shown to stimulate secretome production from murine C2C12 myoblasts while in suspension, whereby conditioning of the bathing media for 30–60 min after PEMF exposure gave the best results [25]. Therefore, in this study, pCM was collected from myoblasts in suspension for 1 h and was employed for the first stage of our preconditioning paradigm (Figure 4, Figure 5, Figure 6), where HTRA1 was uniquely present in the conditioned media (Figure 6B). On the other hand, due to the syncytial complexity of differentiated muscle in vitro, secretome was collected from adherent murine myotube cultures after 6 h of analogous exposure, as 24 h of media conditioning did not enhance the potency (Appendix A). Although murine pCM suppressed the proliferation, colony formation, migration, and invasiveness of both human (MCF-7 and MDA-MB-231) and mouse (67NR and 4T1) breast cancer cell lines, its effect was greatest when administered to murine cancer cell lines. Species specificity hence needs to be taken into consideration when developing therapeutic platforms.

The epithelial–mesenchymal transition (EMT) defines invasive [39] and metastatic [43] cancers, here represented by the MDA-MB-431 and 4T1 cell lines. The MDA-MB-431 (human) and 4T1 (murine) cell lines exhibited higher migratory and invasive capacities than the more epithelial-like cancer cell lines, MCF-7 (human) and 67NR (murine). The pCM suppressed the invasiveness of both the MDA-MB-431 and 4T1 cell lines that also translated to the serum from PEMF exposed mice. Therefore, pCM may potentially provide a source of antimetastatic therapeutics.

The anticancer potential of the myotube pCM was further validated in the vascularized chorioallantoic membrane (CAM) model of the chicken egg [38,40]. The CAM is a validated oncology model with which to evaluate the efficacy of anticancer agents to inhibit tumor growth and angiogenesis [40]. pCM administered to CAM-hosted MCF-7 microtumors stymied their growth and vascularization. The identification of the specific mediators and mechanisms underlying the antiangiogenic effects of pCM has important clinical implications.

### 4.1. Parallels between PEMF-Exposure and Exercise

Mitohormetic parallels between PEMF-exposure and exercise have been hypothesized [3]. The same PEMF paradigm used in the present study was previously shown to preferentially activate skeletal muscle when the whole mouse was exposed [23]. Here, the blood sera collected from either PEMF-exposed or exercised mice were compared for their ability to attenuate breast cancer growth, migration, and invasiveness. Weekly PEMF exposure of mice for 8 weeks (10 min, once weekly) endowed the blood serum with the capacity to reduce cancer growth, migration and invasion when tested in vitro, suggesting that PEMF exposure resulted in skeletal muscle conditioning the blood with anticancer properties. The demonstration of anticancer efficacy from blood collected one week after the last exposure reflects the establishment of mitohormetic adaptations known to be invoked by exercise [3,44]. The in vivo data revealed two general trends. First, the murine serum gave the strongest responses when tested on murine breast cancer cell lines, reconfirming our prior detection of species-specificity. Secondly, the PEMF-serum exhibited stronger anticancer responses than the exercise-serum. Given the ability of PEMF exposure to quickly induce secretome release and its ease of application (10 min exposures to non-invasive magnetic fields) in both cells and animals, magnetic paradigms may represent an easy-to-administer cancer prehabilitative strategy.

Mitohormetic responses comprise adaptations to mitochondrial-derived ROS produced either during the execution of exercise or in response to PEMF exposure [3]. In response to modest oxidative challenge, cells are capable of upregulating their antioxidant defenses, leading to greater resilience and protection against inflammatory damage and cancer, whereas excessive oxidative stress may cause damaging levels of inflammation [3,44]. The tipping point depends on the history of the cell and its potential adaptation to mitochondrial stressors and mitohormesis. In humans, higher exercise intensities are associated with better anticancer outcomes [10,17]. In the present mouse study, the failure of the exercise paradigm to produce an anticancer serum in parity to that achieved with PEMF exposure (Figure 3) could be due to the inadequacy of the exercise protocol, 20 min of aerobic exercise, twice weekly [45]. Alternatively, the blood from exercised mice may have been overly contaminated with inflammatory factors [46]. Although acute inflammation is necessary for muscle repair and adaptation [3], overly strenuous exercise may lead to muscle damage and chronic inflammation [46]. By contrast, the non-invasive and mild PEMF protocol employed in this study has given preliminary indications of possessing anti-inflammatory character [3,25]. Of relevance, an anticancer efficacy of stronger PEMF exposure has been previously demonstrated within behaving mice harboring breast cancer tumors. In this study, weekly exposure of mice to stronger fields (3 mT) for a longer duration (1 h) reduced the size of patient-derived xenographic (PDX) tumors hosted by the mice [38]. Although the mechanism pursued by Tai et al. [38] was different from that investigated here, the degree of muscle crossover invoked by the two distinct exposure paradigms remains to be determined. Given the complicated interrelationship between muscle inflammation and exercise, magnetic exposure may offer a method to invoke anticancer secretome production with a minimum of inflammation and mechanical stress.

The health benefits of regular exercise are manifestations of the actions of the muscle secretome that entrain the metabolic status of the entire organism via a system of muscle tissue secretome crosstalks [3,8,20]. Succinctly, bouts of exercise instigate a cascade of muscular metabolic responses, commencing with muscular mitochondrial activation, catalytically transmitted to the transcriptional apparatus by ROS and oxygen depletion and disseminated to the organism with the mobilization of the muscle secretome [4,20]. The muscle secretome also acts on the muscle itself. Upon being released into the muscle interstitium, the muscle secretome acts via autocrine and paracrine means to modulate muscle development, regeneration, and metabolic and secretory capacities [8]. Given the accepted role of skeletal muscle in establishing whole-body regenerative capacity as well as inflammatory and metabolic statuses, the muscle secretome offers a treasure trove of potential therapeutics to be discovered.

### 4.2. pCM Preconditioning System for Enhanced Secretome Characteristics

The present study demonstrates that the preconditioning of differentiating myoblasts with pCM later (as myotubes) accentuated their constitutive secretome release (Figure 4B; light grey box) as well as augmented PEMF-induced secretome release (Figure 4B; dark grey box). Notably, the collection of constitutively released secretome from progenitor cells for regenerative medicine applications is a standard protocol and would be represented in our preconditioning paradigm by the basal media preconditioning condition (Grey Triangles, Figure 6E), representing an inferior secretome collection scenario. Key features contributing to the efficacy of this preconditioning paradigm were the collection of pCM from myoblasts while in suspension and the application of the PEMFs to the muscle cells in the downward direction (Figure 4, Figure 6 and Appendix A). Meeting these two criteria greatly enhanced the potency of the secretome.

The consequences of pCM preconditioning were most noticeable in experiments examining the human breast cancer cell lines that demonstrated a novel cross-species sensitivity to murine constitutively released secretome from preconditioned myotubes (Figure 5; middle blue bars and squares), compared to a relative insensitivity to the murine constitutively released secretome derived from “nonpreconditioned” myotubes (Figure 1; middle red bars and squares). Preconditioning differentiating muscle cells with pCM was hence capable of enhancing the anticancer potency of the constitutively released secretome. The preconditioning paradigm was molecularly recapitulated upon examining the HTRA1 expression pattern. HTRA1 was synthesized and released upon PEMF exposure, promoting muscle differentiation that, in turn, further augmented secretome release, and thereby reinforced the anticancer response (Figure 6). In essence, HTRA1 is required for muscle maturation and function, of which, anticancer defense is part. Our in vitro preconditioning paradigm demonstrates mitohormetic adaptations similar to those invoked in vivo by exercise training. Accordingly, HTRA1 levels were elevated in the serum of both exercised and PEMF-exposed mice (Figure 6I). The presented preconditioning paradigm may also ultimately serve as the basis for the development of molecular therapeutics platforms exploiting the secretome of muscle or other progenitor cell classes.

### 4.3. HTRA1 Reinforces Muscular Anticancer Response

HTRA1 was previously shown to be upregulated in C2C12 muscle cells by direct PEMF exposure as well as in response to the administration of pCM [25]. HTRA1 was also shown necessary for muscle differentiation (Figure 6D) [28,37] that correlated with greater HTRA1 secretion from myotubes (Figure 6A and Appendix A) [29]. Preconditioning myoblasts with pCM further enhanced myogenic differentiation (Figure 6E) as well as HTRA1 expression (Figure 6E(v)). The designed preconditioning strategy exploits the ability of the muscle secretome to promote the differentiation of muscle, which, in turn, strengthens the secretome response.

Muscle health is important for human resilience to cancer and is an area of active research [4,8]. Demonstrating anticancer necessity and sufficiency, recombinant HTRA1 attenuated MCF-7 growth (Figure 6G), whereas depleting HTRA1 from pCM abolished anticancer potency (Figure 6H). These findings align with the anticancer capacity of pCM and independently validate our general secretome findings (Figure 1, Figure 2, Figure 4 and Figure 5). Muscle-secreted HTRA1 modulates cancer progression [29] by attenuating TGF-β signaling required for angiogenesis [47]. The presence of HTRA1 in the pCM may account for the suppression of tumor vascularization observed in the CAM-hosted MCF-7 microtumors (Figure 2). HTRA1 levels were also shown to be upregulated in the blood of PEMF-exposed and exercised mice (Figure 6I), aligning with the efficacy of these sera to attenuate cancer growth, migration, and invasiveness (Figure 3). An inverse relationship exists between muscle maintenance and cancer survival [48]. Given that HTRA1 levels are reduced in several forms of cancer [29], in conjunction with data reported here and elsewhere demonstrating a role for HTRA1 in muscle development, the induction of HTRA1 by low-energy PEMF exposure assumes clinical relevance in the realm of cancer management on several levels.

In the present study, focus was placed on HTRA1 action over other myokines to expand on an earlier report [25] of HTRA1 induction with the same magnetic platform technology, as well as to provide a molecular foundation for our debuted preconditioning paradigm. This is not to imply that other myokines are not contributing to the adaptive developmental response to the presented preconditioning system. Our future studies will focus on identifying and characterizing other known and novel secretome agents that are similarly upregulated by PEMF exposure and lead to muscle secretome adaptation, as well as examining the extracellular vesicle component of the muscle secretome for clinical and regenerative medicine applications.

## 5. Conclusions

Muscle cells briefly exposed to PEMFs produced a conditioned media capable of inhibiting breast cancer cell growth, migration, and invasion in vitro and ex vivo, whereas conditioned media harvested from unexposed muscle cells, representing the constitutive release of secretome from unexposed muscle cells, was less effective. Blood sera collected from PEMF-exposed or exercised mice also inhibited the proliferation, migration, and invasiveness of breast cancer cells, providing physiological relevance to the in vitro results. A secretome preconditioning methodology was designed to enhance secretome anticancer potency. Importantly, the anticancer potency of the constitutively released secretome from unexposed muscle was also enhanced. Myogenic differentiation was also enhanced with pCM preconditioning, confirming an adaptive response to secretome induction akin to exercise training in animals. Via this drug-free system, secretome production and release are both effectively enhanced with minimal non-invasive intervention. Given the demonstrated ability of PEMF exposure to induce secretome release and its ease of application (10 min exposures) in both cells (once) and animals (weekly), magnetic paradigms may offer a unique opportunity for soluble biomarker discovery and clinical exploitation. Aside from the immediate clinical implications of muscle secretome magnetic induction, this study paves the way for the development of drug-free magnetic platforms for the discovery and characterization of novel liquid biomarkers with relevance to human cancer; HTRA1 was highlighted as a notable example.

## Figures and Tables

**Figure 1 cells-13-00460-f001:**
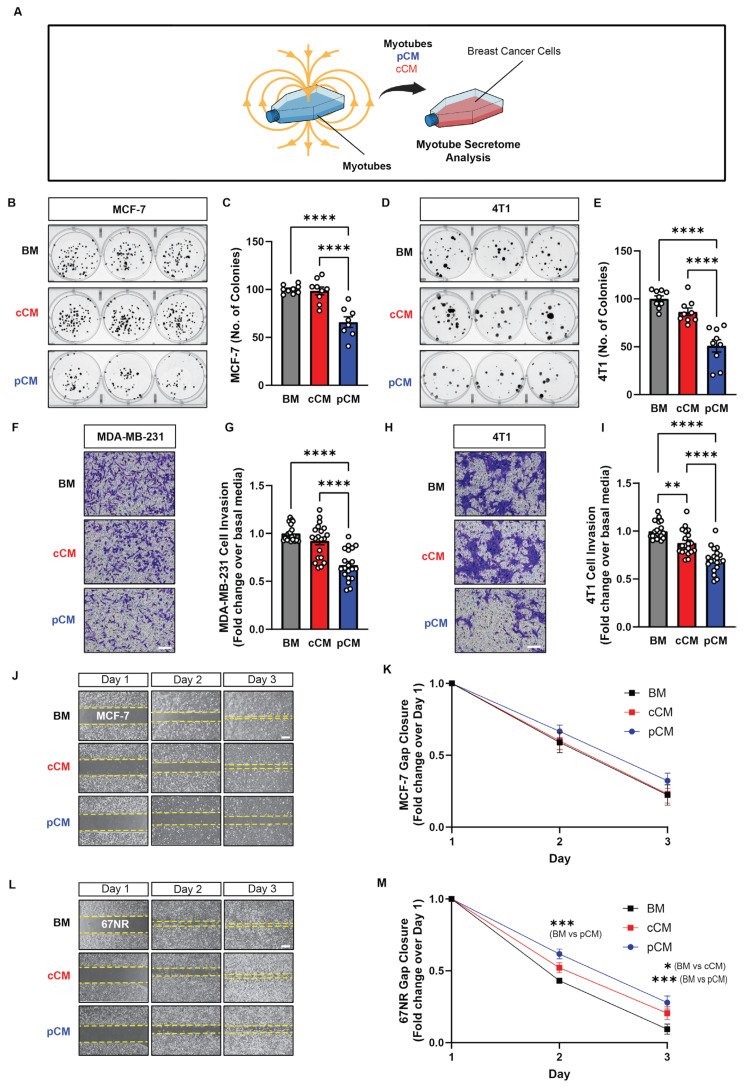
Effect of magnetically induced muscle secretome on breast cancer cell growth, invasion, and migration. (**A**) Schematic of the experimental paradigm. (**B**) MCF-7 cell colony formation following the administration of basal media (BM; grey), control conditioned media (cCM; red), or PEMF conditioned media (pCM; blue) 24 h after cell-seeding at clonal density. Myotubes were allowed to condition BM (DMEM supplemented with 5% FBS) for 6 h following PEMF exposure (pCM), or not (cCM). (**C**) Quantification of colony number after 11 days of treatment with the indicated media (*n* = 3). (**D**) 4T1 cell colony formation following the administration of the indicated media and the corresponding quantification at day 7 (**E**) (*n* = 3). (**F**) Representative images of invading (blue-stained) MDA-MB-231 cancer cells 48 h following treatment in the indicated media. (**G**) Fold change in invading MDA-MB-231 cells normalized to the BM scenario (*n* = 5). (**H**) Representative images of invading (blue-stained) 4T1 cancer cells 48 h following treatment in the indicated media. (**I**) Fold change in invading 4T1 cells normalized to the BM scenario (*n* = 5). (**J**) Brightfield images of gap closure at days 1, 2, and 3 after the provision of the indicated media 24 h after the plating of MCF-7 cells. (**K**) Fold change in gap closure for MCF-7 cells over 3 days in response to the indicated media (*n* = 5). (**L**) Brightfield images of gap closure at days 1, 2, and 3 after the provision of the indicated media 24 h after the plating of 67NR cells. (**M**) Fold change in gap closure for 67NR cells over 3 days in response to the indicated media (*n* = 5). Statistical analyses were performed on minimally 4 independent biological replicates, with * *p* < 0.05, ** *p* < 0.01, *** *p* < 0.001, and **** *p* < 0.0001. Scale bar = 300 µm. The error bars represent the standard error of the mean. Dots represent all technical replicates from all biological replicates.

**Figure 2 cells-13-00460-f002:**
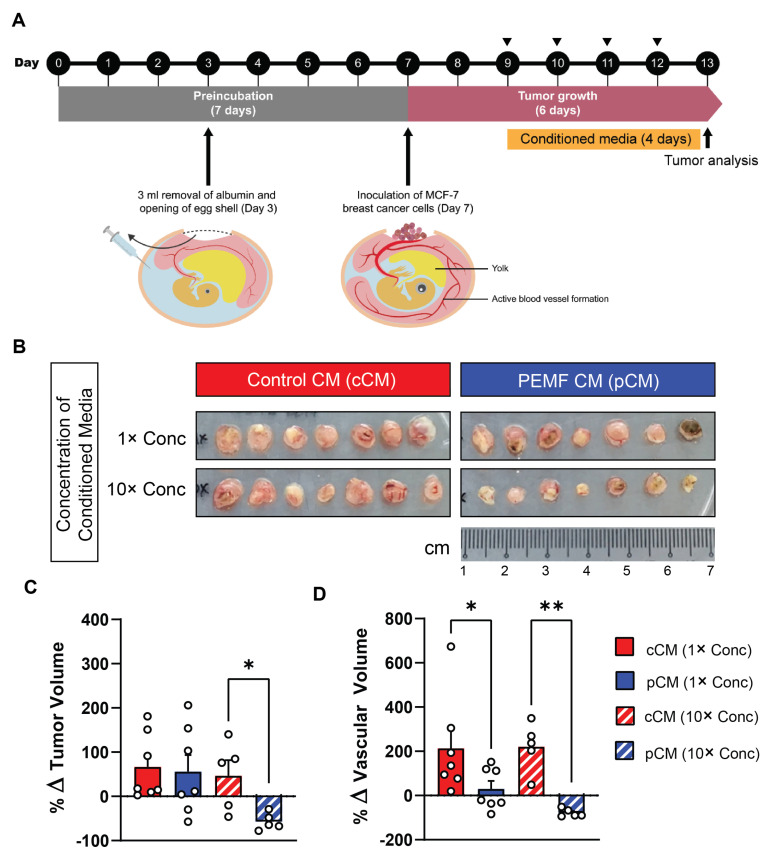
Effect of muscle pCM on the growth and vascularization of breast cancer microtumors engrafted onto the chorioallantoic membrane (CAM) of the chicken egg. (**A**) Schematic of experimental protocol. Conditioned media (CM) were provided to MCF-7 microtumors in the CAM model. The daily provision of CM is indicated by the black triangles. (**B**) Representative images of isolated MCF-7 microtumors on day 7 after exposure to either 1× or 10× concentrated pCM or cCM. (**C**) Quantification of the % change in tumor volume for the indicated conditions. (**D**) Quantification of tumor vascularity expressed as % change in vascular volume. Statistical analyses were performed on 5–7 independent CAM, with * *p* < 0.05 and ** *p* < 0.01. The error bars represent the standard error of the mean. Dots represent individual biological replicates.

**Figure 3 cells-13-00460-f003:**
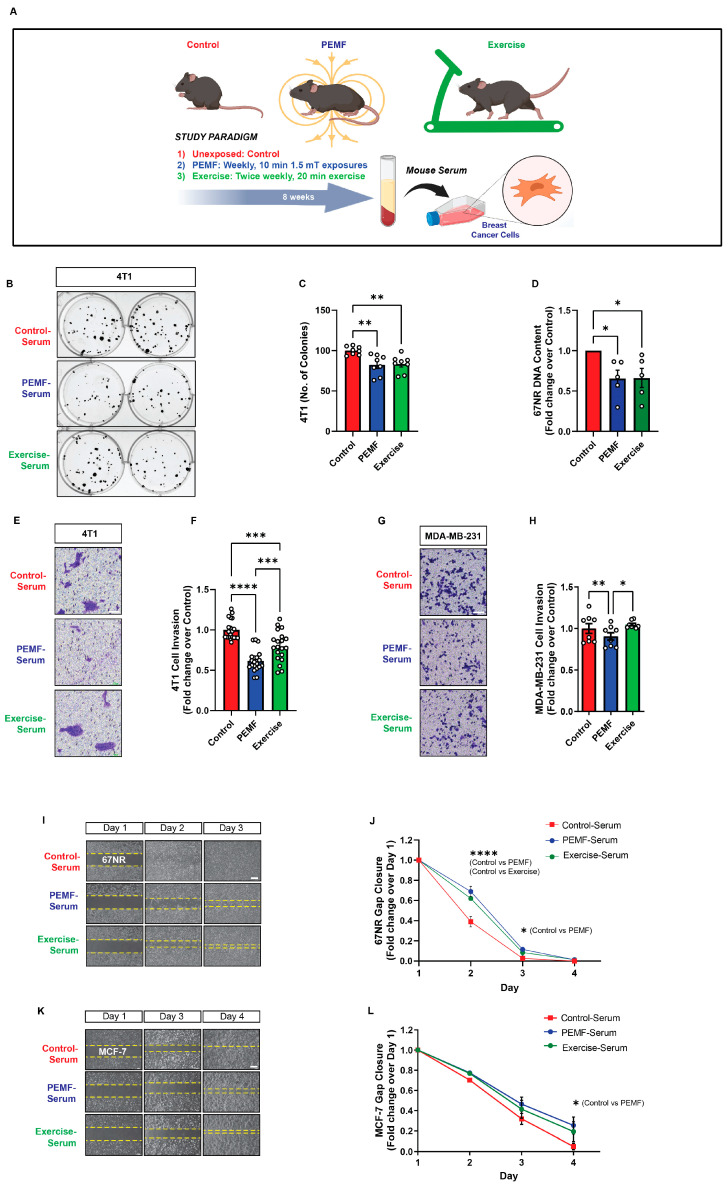
Effect of serum from magnetically exposed mice on the migration and invasion of breast cancer cells. (**A**) Schematic of PEMF and exercise protocols. C57BL/6 mice were either exposed to PEMFs for 10 min once per week or exercised on a treadmill twice weekly for 20 min for 8 weeks. Blood was collected one week after the last intervention. Breast cancer cells were provided with serum from control (red), PEMF-exposed (blue), or exercised (green) mice. (**B**) Cell colony formation in the presence of the various mice sera 24 h after the seeding of 4T1 cells at clonal density. (**C**) Quantification of colony number on day 7 (*n* = 4). (**D**) Quantification of 67NR cell DNA content 24 h after the provision of the indicated sera. Data are expressed as fold change over control-serum and the dots represent the average of independent replicates (*n* = 5). (**E**) Representative images of invading (blue-stained) 4T1 cancer cells 48 h following treatment in the indicated mice sera. (**F**) Fold change in the invading 4T1 cells normalized to the control-serum scenario (*n* = 5). (**G**) Representative images of invading (blue-stained) MDA-MB-231 breast cancer cells 48 h following treatment with the indicated sera. (**H**) Fold change in invading MDA-MB-231 cells normalized to the control-serum scenario (*n* = 2 independent biological replicates of two mice each). (**I**) Brightfield images showing gap closure over 4 days following the administration of the indicated mice sera 24 h after the plating of 67NR cells. (**J**) Fold change in 67NR cell gap closure in response to the indicated mice sera over 4 days (*n* = 5). (**K**) Brightfield images showing cell gap closure over 4 days following the administration of the indicated mice sera 24 h after the plating of MCF-7 cells. (**L**) Fold change in MCF-7 cell gap closure in response to the indicated mice sera over 4 days (*n* = 2 independent biological replicates of two mice each). Statistical analyses were performed minimally in three independent biological replicates with * *p* < 0.05, ** *p* < 0.01, *** *p* < 0.001, and **** *p* < 0.0001. The error bars represent the standard error of the mean. Unless otherwise stated, dots represent all technical replicates from all biological replicates. Scale bar = 300 µm. Illustration in (**A**) was created with BioRender.com, (accessed on 31 August 2023).

**Figure 4 cells-13-00460-f004:**
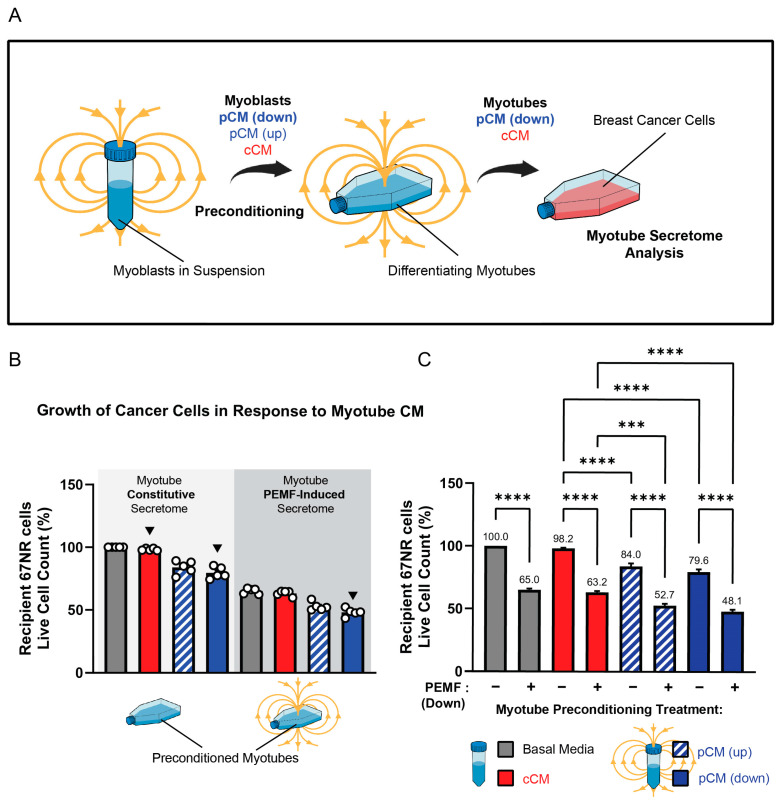
Effect of pCM preconditioning on breast cancer cell proliferation. (**A**) Schematic of the preconditioning paradigm. (**B**) Secretome-modulated 67NR breast cancer cell growth. Secretomes were collected from myotubes in response to the indicated combinations of pCM preconditioning and direct PEMF exposure (dark grey), or not (light grey). Cancer cell quantification was conducted 24 h after the provision of the indicated myotube-conditioned media (see Panel (**A**)). All presented values were normalized to the response of secretome collected from myotubes differentiated in fresh (unconditioned) basal media ((**B**), first grey bar). (**C**) Comparison of each preconditioning paradigm to itself with reference to PEMF exposure (+), or not (−). Numbers over each histogram indicate the % change relative to the unexposed basal media response. Basal media consisted of fresh DMEM (no FBS) and all conditionings were of fresh DMEM (no FBS). Statistical analyses were performed in five independent biological replicates, each representing the mean of three technical replicates, with *** *p* < 0.001, and **** *p* < 0.0001. The error bars represent the standard error of the mean. Each dot represents the average of three technical replicates originating from five independent biological replicates.

**Figure 5 cells-13-00460-f005:**
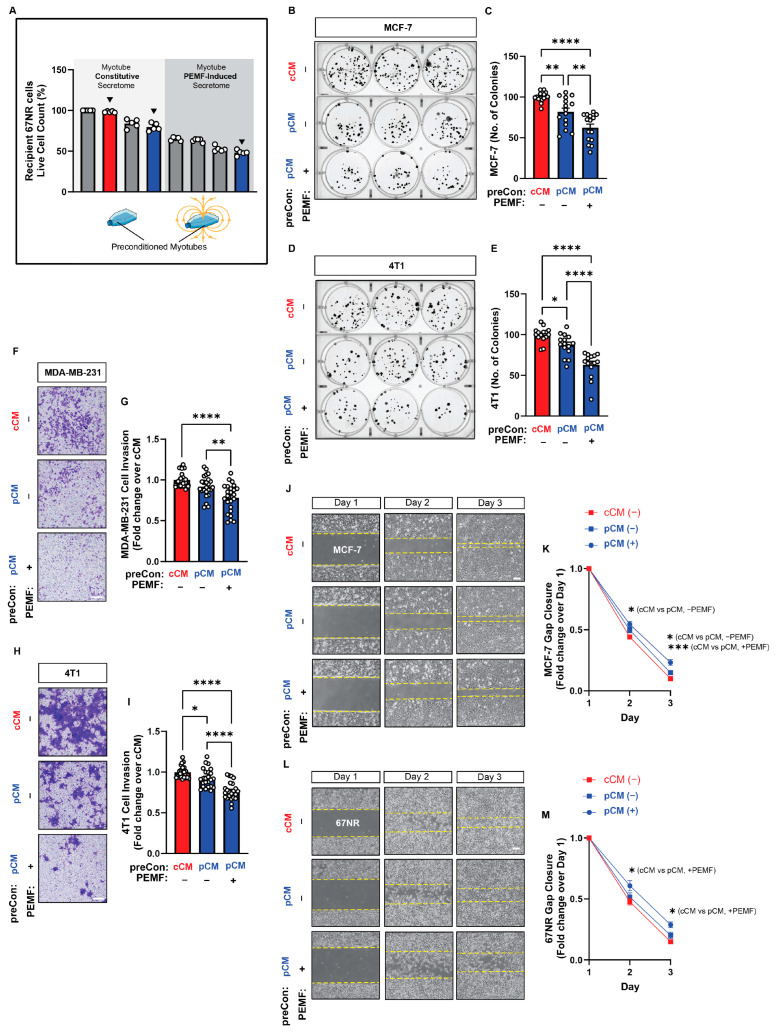
Effect of pCM preconditioning on breast cancer cell colony formation, invasion, and migration. (**A**) Schema depicting the conditions of Figure 4 (black triangles) tested under results presented in Figure 5. (**B**) MCF-7 colony formation in response to myotube secretome. Myotubes were first preconditioned with conditioned media harvested from either unexposed (cCM; red) or PEMF-exposed (pCM; blue) myoblasts. Subsequently, only myotubes preconditioned with pCM were exposed (+ PEMF) or not (-−PEMF) before secretome collection for testing as indicated. The cCM condition was used as a reference scenario. The conditions analyzed correspond to the first (red bar), second (middle blue bar), and third (end blue bar) triangles in Figure 4B. (**C**) Quantification of surviving colonies (*n* = 5). (**D**) Colony formation of 4T1 cells following exposure to myotube secretome under the indicated preconditioning treatments. (**E**) Quantification of surviving colonies (*n* = 5). (**F**) Representative images of invading (blue-stained) MDA-MB-231 cancer cells 48 h following treatment in the indicated myotube-conditioned media. (**G**) Fold change in invading MDA-MB-231 cells normalized to the cCM scenario (*n* = 6). (**H**) Representative images of invading (blue-stained) 4T1 cancer cells 48 h following treatment in the indicated conditioned media. (**I**) Fold change in invading 4T1 cells normalized to the cCM scenario (*n* = 6). (**J**) Brightfield images showing cell gap closure over 3 days following the administration of the indicated conditioned media 24 h after the plating of MCF-7 cells. (**K**) Fold change in gap closure in response to the indicated conditioned media over 3 days (*n* = 5). (**L**) Brightfield images showing cell gap closure over 3 days following the administration of the indicated conditioned media 24 h after the plating of 67NR cells. (**M**) Fold change decreasing in gap distance in response to the indicated conditioned media over 3 days (*n* = 5). All PEMF exposures were in the downward direction. Statistical analyses were performed on four independent biological replicates with technical replications, with * *p* < 0.05, ** *p* < 0.01, *** *p* < 0.001, and **** *p* < 0.0001. Scale bar = 300 µm. The error bars represent the standard error of the mean. Dots represent all technical replicates from all biological replicates.

**Figure 6 cells-13-00460-f006:**
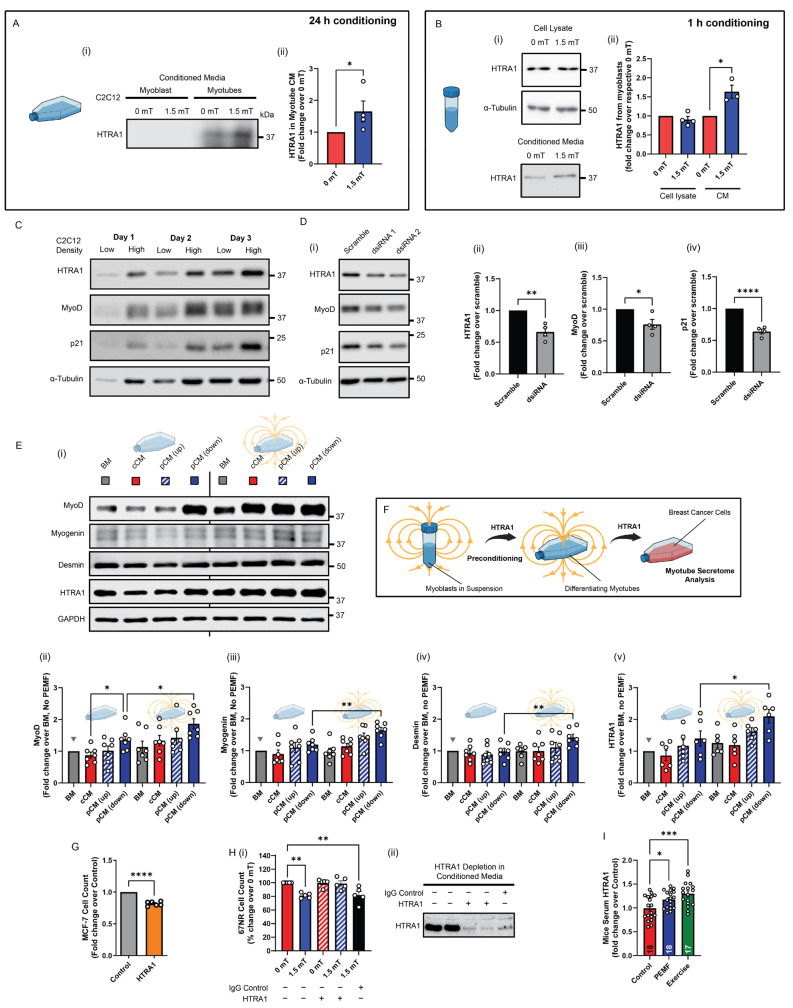
HTRA1 recapitulates pCM preconditioning response. (**A**(**i**)) Immunoblot showing the relative abundance of HTRA1 (37 kDA) detected in myotubes from conditioned media (CM) 24 h post PEMF exposure. (**A**(**ii**)) Quantification of HTRA1 in myotubes adherent cultures expressed as fold change relative to respective 0 mT controls. HTRA1 (37 kDa) was not readily detected in CM from myoblasts adherent cultures (*n* = 4). (**B**(**i**)) Representative Western blot and the corresponding quantification (**ii**) of HTRA1 levels in cell lysates and CM of myoblasts in suspension exposed to PEMFs, or not, as indicated. CM was collected 1 h post PEMF exposure (*n* = 4). (**C**) Representative Western blots showing the expression of HTRA1, MyoD, and p21 in myoblasts under low- and high-density seeding, over 3 days in culture. (**D**(**i**)) Immunoblots showing the knockdown of HTRA1 in C2C12 myoblasts using two independent dsiRNA. The corresponding bar charts show the mean relative fold change pooled from two independent dsiRNA in protein expression of (**ii**) HTRA1, (**iii**) MyoD, and (**iv**) p21 after silencing of HTRA1 (*n* = 2 per dsiRNA). (**E**(**i**)) Representative blots and their corresponding pooled data showing the expression of myogenic markers (**ii**) MyoD, (**iii**) myogenin, (**iv**) desmin, and (**v**) HTRA1 from preconditioned myotubes exposed to PEMFs, or not (*n* = 6). (**F**) Schematic depiction of how pCM/HTRA1 preconditions proliferating myoblasts to enhance differentiation the anticancer secretome response of the resultant myotubes (*n* = 6). (**G**) Bar chart shows the relative live cell count of MCF-7 breast cancer cells 24 h after treatment with recombinant HTRA1 (10 nM) (*n* = 5). (**H**(**i**)) Bar chart depicting 67NR relative percentage cell count when given HTRA1-depleted CM (hatched red and blue) or not. Black bar corresponds to the control scenario of 1.5 mT-stimulated CM in the presence of non-specific IgG. (**ii**) Representative western blot showing the depletion of HTRA1 protein using antibody depletion. (**I**) Bar chart showing the relative fold change in HTRA1 measured using ELISA from mice blood serum. The number of independent animals is depicted within each bar (*n* = 17–18). Statistical analyses were performed on minimally three independent biological replicates, with * *p* < 0.05, ** *p* < 0.01, *** *p* < 0.001, and **** *p* < 0.0001. The error bars represent the standard error of the mean, and each dot represents independent experiments. Also see Appendix A.

**Table 1 cells-13-00460-t001:** List of primary antibodies used for Western blot analysis.

Antibody Name	Dilution Factor	Cat. No.	Manufacturer
HTRA1	1:1000	#55011-1-AP	Proteintech
MyoD	1:300	#sc-71629	Santa Cruz
Myogenin	1:300	#sc-12732	Santa Cruz
p21	1:300	#sc-6246	Santa Cruz
Desmin	1:300	#ab907	Merck
GAPDH	1:10,000	#60004-1-1g	Proteintech
α-tubulin	1:10,000	#66031-1-1g	Proteintech
β-actin	1:10,000	#60008-1-1g	Proteintech

## Data Availability

All data supporting the results are presented in the manuscript. Any other inquiries can be directed to the corresponding authors via email.

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
