# Peer review of "Secretome from Magnetically Stimulated Muscle Exhibits Anticancer Potency: Novel Preconditioning Methodology Highlighting HTRA1 Action"

_cells, 2024, doi:10.3390/cells13050460_

Round 1
Reviewer 1 Report
Comments and Suggestions for Authors
In this manuscript authors found that a brief exposure to low amplitude PEMFs generated a conditioned media able to mitigate breast cancer cell growth, migration, and invasiveness. Moreover, administering of this conditioned media to breast cancer microtumors engrafted onto the chorioallantoic membrane of chicken eggs reduced tumor volume and vascularity. Authors also demonstrated that the serine protease HTRA1 was shown to be upregulated in the conditioned media and was necessary for the anticancer potency of the media itself. Moreover, recombinant HTRA1 added to basal media recapitulated the anticancer effects of the conditioned media while antibody-based absorption of HTRA1 from the conditioned media precluded its anticancer effects.
The manuscript is interesting, generally well written and appropriately ilòlustrated. Figures are clear. Only some points should be improved.
Minor points:
Lines 101-104: reference 29 does not talk about prostate cancer
Lines 108-110: It deserves to be mentioned that the secret form of HTRA1 shows an inverted correlation with tissue expression. In fact, it deserves to be pointed out that the secreted HTRA1 levels are significantly increased in inflammatory diseases (see PMID: 31536940, 28585581 and 35131488). This is an important point to add since it highlight HTRA1 also as an inflammatory marker.
2.3. Cancer Cell Count, Colony Formation, Invasion and Migration Assays: these methods must be fully described
2.7. Western blot and Silencing of HTRA1: The primary antibodies used should be inserted in a dedicate table
Please, add the number of replicates (N) in the legend of each figure
Abbreviations must be written in full length when mentioned for the first time
Major points:
there are no major points to highlight since study's methodology, technique and experimental controls are appropriated.
Reviewer 2 Report
Comments and Suggestions for Authors
Previous findings have been indicating that exercise may affect the progression of malignant diseases in-vivo. There is therefore a rational justification to further study and examine the ability of muscle cells to secret factors that would restrain cancer development in human. Along this notion, in the current manuscript the authors show that conditioned media from myotubes subjected to low amplitude pulsed electromagnetic field (PEMF) contain cell released secrotome that can attenuate cancer cells growth both in-vitro and in an in-vivo model. Furthermore, this secretome is also capable of attenuating the migration and invasiveness of metastatic cancer cells. Importantly, the authors show that exercised muscles in mice can also to release anti-cancer sceretome and that the main active component in the PEMF stimulated myotubes, is the serine protease -HTRA1. Obviously, deciphering the mechanism of action of HTRA1 would endow this manuscript with another scale of significance. Overall, this work extends our knowledge of the link between muscles stimulation and release of anti-cancer factors, a phenomenon that has also potential translational ramifications.
However before approving this manuscript for publication, the authors should address the following points:
1. To document the specificity of the PEMF stimulated myotubes secrotome activity, it would be beneficial to examine also its effects on the growth of normal human cells (epithelial or fibroblastic cells).
2. From the data presented in the manuscript it is not clear whether the myotubes stimulated secretome affects primarily the survival (meaning that it evokes cell death), or whether it affects also the replication (by affecting the cell-cycle), of the treated malignant cells. Therefore, in Figure 1 C and E, Figure 3 C , and Figure 5 C,E, the authors should write “number of colonies”, rather then (%) Survival.
